# Antioxidative Effects of Chrysoeriol via Activation of the Nrf2 Signaling Pathway and Modulation of Mitochondrial Function

**DOI:** 10.3390/molecules26020313

**Published:** 2021-01-09

**Authors:** Myung Hee Kim, So Yeon Kwon, So-Yeun Woo, Woo Duck Seo, Dae Yu Kim

**Affiliations:** 1Inha Research Institute for Aerospace Medicine, Inha University, Incheon 22212, Korea; 318560@inha.ac.kr; 2Department of Mechanical Engineering, College of Engineering, Inha University, Incheon 22212, Korea; 12180914@inha.edu; 3Rural Development Administration, National Institute of Crop Science, Wanju-gun, Jeollabuk-do 55365, Korea; soyeon@korea.kr (S.-Y.W.); swd2002@korea.kr (W.D.S.); 4Department of Electrical Engineering and Center for Sensor Systems, College of Engineering, Inha University, Incheon 22212, Korea

**Keywords:** chrysoeriol, antioxidants, mitochondrial function, retinal pigment epithelium, age-related macular degeneration

## Abstract

Retinal pigment epithelium (RPE) cell dysfunction caused by excessive oxidative damage is partly involved in age-related macular degeneration, which is among the leading causes of visual impairment in elderly people. Here, we investigated the protective role of chrysoeriol against hydrogen peroxide (H_2_O_2_)-induced oxidative stress in RPE cells. The cellular viability, reactive oxygen species (ROS) generation, and mitochondrial function of retinal ARPE-19 cells were monitored under oxidative stress or pre-treatment with chrysoeriol. The expression levels of mitochondrial-related genes and associated transcription factors were assessed using reverse transcription–quantitative polymerase chain reaction (RT-qPCR). Moreover, the protein expression of antioxidant signal molecules was characterized by Western blot analysis. Chrysoeriol significantly increased cell viability, reduced ROS generation, and increased the occurrence of antioxidant molecules in H_2_O_2_-treated ARPE-19 cells. Additionally, mitochondrial dysfunction caused by H_2_O_2_-induced oxidative stress was also considerably diminished by chrysoeriol treatment, which reduced the mitochondrial membrane potential (MMP) and upregulated mitochondrial-associated genes and proteins. Chrysoeriol also markedly enhanced key transcription factors (Nrf2) and antioxidant-associated genes (particularly HO-1 and NQO-1). Therefore, our study confirms the protective effect of chrysoeriol against H_2_O_2_-induced oxidative stress in RPE cells, thus confirming that it may prevent mitochondrial dysfunction by upregulating antioxidant-related molecules.

## 1. Introduction

Age-related macular degeneration (AMD) is a serious retinal disease that causes irreversible loss of central visual function primarily in the elderly populations of developed countries. AMD is classified into two major forms—a non-neovascular dry form and a neovascular exudative wet form [1,2]. Dry AMD (atrophy-associated AMD) causes alterations in the pigment cell layer, which contains the retinal pigment epithelium (RPE) and sub-retinal sediments, due to lipid and protein accumulation between these cells and Bruch’s membrane (also known as drusen). These processes eventually result in RPE cell death, photoreceptor malfunction, and central vision loss [1,2,3]. Currently, anti-vascular endothelial growth factor (anti-VEGF) therapy has been successfully used to treat wet AMD; however, effective treatments for dry AMD are not available yet [4,5]

Although the etiological mechanisms of dry AMD are not known, patients with this condition exhibit elevated oxidative stress and inflammation in the RPE [5,6,7]. RPE cells exhibit high metabolic rates with enriched mitochondrial populations, and the oxidative phosphorylation processes associated with mitochondrial function produce adenosine triphosphate (ATP), which in turn leads to the generation of high amounts of reactive oxygen species (ROS) [8] Thus, RPE cells are constantly in contact with abundant endogenous ROS. Long-term accumulation of oxidative damage leads to RPE cell dysfunction and increases the cellular susceptibility to oxidative stress. ROS predominantly target mitochondria and compromise their membrane integrity, consequently disrupting the mitochondrial membrane potential (ΔΨm, MMP), causing mitochondrial dysfunction, and jeopardizing the survival of the cell. Therefore, intramitochondrial oxidative stress is closely linked to cell survival and other related processes including mitochondrial flexibility, apoptosis, and autophagy in AMD [9]. Thus, protecting RPE cells from oxidative damage is crucial to prevent AMD.

Flavonoids, a group of polyphenol compounds, are a well-characterized major class of phytochemicals. Herbal medicine has traditionally been used to improve metabolism, and several compounds have been reported to possess antioxidant and anti-inflammatory properties [10]. Flavonoids are widely found in nature and common food items. Thus, these compounds can be used as dietary supplements for disease prevention. Chrysoeriol (5,7-dihydroxy-2-(4-hydroxy-3-methoxyphenyl)chromen-4-one) (Figure 1A, CHE) is a flavonoid compound that is commonly found in plants of the genus *Perilla frutescens*. This compound has been reported to possess several health-beneficial properties, including antioxidant [11,12,13], anti-inflammatory [14,15], anti-tumor [16,17,18], anti-osteoporosis [19], and hepatoprotective [20] properties. However, the antioxidant effects of chrysoeriol on RPE cells in AMD patients and its potential to regulate mitochondrial function remain unknown. Therefore, our study sought to investigate the effects of chrysoeriol on H_2_O_2_-induced oxidative stress, mitochondrial dysfunction, and cell death in ARPE-19 cells. Our findings show that chrysoeriol can protect RPE cells from H_2_O_2_-induced oxidative stress and associated mitochondrial dysfunction by upregulating antioxidant proteins and modulating mitochondrial dynamics.

## 2. Results

### 2.1. Chrysoeriol Protected ARPE-19 Cells against H_2_O_2_-Induced Death

To evaluate the optimal concentration of chrysoeriol (Figure 1A) to be used without causing cytotoxicity, ARPE-19 cells were incubated with various concentrations of chrysoeriol for 24, 48, and 72 h. As shown in Figure 1B, chrysoeriol (2.5–10 μM) did not exert any visible cytotoxic effect on ARPE-19 cells compared with the control group. In contrast, chrysoeriol concentrations between 20 and 40 μM decreased the cell viability after 48 and 72 h. Therefore, chrysoeriol was administered at the concentrations of 2.5 and 5 μM in the subsequent experiments. To determine a suitable H_2_O_2_ concentration for the induction of oxidative damage, the cells were exposed to various doses of H_2_O_2_ for 24 h. H_2_O_2_ significantly reduced the cell viability to approximately 55% when used at 500 μM, and thus this concentration was utilized in the subsequent experiments (IC 50; 528 ± 4.2 μM, Figure 1C). To test the protective effect of chrysoeriol on H_2_O_2_-induced cell death, the cells were pre-treated with chrysoeriol for 2 h before being exposed to H_2_O_2_ for 24 h. As illustrated in Figure 1D, the pre-treatment with chrysoeriol significantly prevented the H_2_O_2_-induced death of ARPE-19 cells almost as much as the pre-treatment with *N*-acetyl-cysteine (NAC), used as an antioxidant control.

### 2.2. Chrysoeriol Suppressed H_2_O_2_-Induced Oxidative Stress in ARPE-19 Cells

Excessive ROS accumulation is considered to be one of the primary sources of cell damage. Using the fluorescence probes 2,7-dichlorofluorescein diacetate (H_2_DCFDA) and dihydroethidium (DHE), the intracellular ROS levels in H_2_O_2_-treated ARPE-19 cells were quantitated. As shown in Figure 2, 500 μM H_2_O_2_ significantly increased the fluorescent intensities of H_2_DCFDA (A) and DHE (B) in the ARPE-19 cells compared with the control group. However, pre-treating ARPE-19 cells with chrysoeriol substantially decreased the ROS levels compared with the levels in the cells treated with H_2_O_2_ alone (Figure 2A,B). The suppressive activity of chrysoeriol was also evident in the H_2_DCFDA fluorescence image (Figure 2C). Additionally, the ROS levels in the cells treated with chrysoeriol alone were not significantly different than the levels in untreated cells. To examine the potential protective properties of chrysoeriol against oxidative damage, the expression levels of genes encoding the antioxidant proteins SOD1, SOD2, catalase (CAT), and GPx were assessed using reverse transcription–quantitative polymerase chain reaction (RT-qPCR) (Figure 3A). According to our gene expression analyses, chrysoeriol pre-treatment significantly increased the expression of antioxidant genes (particularly SOD2 and GPx) compared with the expression levels in the cells treated with H_2_O_2_ alone. Additionally, chrysoeriol pre-treatment, versus H_2_O_2_ treatment alone, markedly upregulated HO-1, NQO-1, and Nrf2 expression (Figure 3B). Moreover, as shown in Figure 3C, pre-treatment with chrysoeriol markedly enhanced Nrf2 protein expression levels, which was largely abrogated by treatment with H_2_O_2_. Although the expression of HO-1 and NQO-1 was lower than that of Nrf2, these genes exhibited the same protein expression trends as Nrf2 (Figure 3C,D). Interestingly, Nrf2 protein level significantly increased by chrysoeriol treatment alone in normal ARPE-19 cells. These results suggest that chrysoeriol is a potent activator of Nrf2 expression in ARPE-19 cells.

### 2.3. Chrysoeriol Enhanced Mitochondrial Function by Upregulating Mitochondrion-Related Genes

Growing evidence has linked the activation of Nrf2 expression to mitochondrial function and integrity under stressful conditions [21,22,23]. Therefore, to further determine whether chrysoeriol is involved in mitochondrial function, the mitochondrial membrane potential (MMP, ∆Ψm) of ARPE-19 cells was characterized by analyzing their red/green fluorescence intensity ratio via the JC-10 assay. As shown in Figure 4A, the ARPE-19 cells exposed to 500 μM H_2_O_2_ exhibited a reduced red/green fluorescence intensity ratio, indicating ∆Ψm dissipation, as in the cells treated with carbonyl cyanide m-chlorophenyl hydrazone (CCCP; i.e., a mitochondrial oxidative phosphorylation uncoupler). However, pre-treatment with chrysoeriol at 2.5 or 5 μM for 2 h improved the H_2_O_2_-induced ∆Ψm reduction similarly to the antioxidant positive control (NAC) (Figure 4A). Additionally, chrysoeriol alone did not have a significant effect on the MMP of ARPE-19 cells compared with the control cells. To understand the underlying mechanisms by which chrysoeriol protects mitochondrial function, the expression of mitochondrial respiration and mitochondrial dynamics genes was studied via RT-qPCR. As illustrated in Figure 4B, the mRNA levels of mitochondrial transcription factors (transcription factor A, mitochondrial TFAM) and DNA replication genes (polymerase (DNA-directed), gamma (POLG)) were significantly increased by chrysoeriol pre-treatment compared with H_2_O_2_-alone treatment. Moreover, the expression levels of oxidative phosphorylation (OXPHOS)-associated genes, including ATP synthase subunit O (ATP5O), COX4I1 (cytochrome c oxidase subunit 4 isoform 1), cytochrome c oxidase subunit 5B (COX5b), and NADH dehydrogenase (ubiquinone) 1 beta subcomplex 5 (NDUFB5), were significantly increased by chrysoeriol pre-treatment compared with H_2_O_2_-alone treatment or DMSO treatment (Figure 4C). Moreover, the expression levels of genes related to mitochondrial dynamics, such as fission 1 (FIS1) and mitofusin 1 and 2 (MFN1 and 2), were significantly upregulated in the cells pre-treated with chrysoeriol compared with the levels in H_2_O_2_-alone-treated or control cells (Figure 4D). These results suggest that the protective effect of chrysoeriol against H_2_O_2_-induced cell damage in ARPE-19 cells is mediated through the upregulation of mitochondrial biogenesis genes.

### 2.4. Chrysoeriol Regulated Mitochondrial Process Proteins

Oxidative stress is one of the major contributors to mitochondrial function, due to its modulatory effects on mitochondrial fusion and fission dynamics [24,25,26]. To further characterize the protective role of chrysoeriol in mitochondrial processes triggered by H_2_O_2,_ Western blot analyses were conducted to quantify the levels of crucial mitochondrial proteins. As shown in Figure 5, when ARPE-19 cells were exposed to chrysoeriol, TOM-20 (i.e., an outer mitochondrial membrane (OMM) marker) and MFN2 were significantly upregulated compared with the levels in cells treated with H_2_O_2_ alone or in the standard control. Interestingly, 5 µM chrysoeriol treatment alone upregulated TOM20 but not MFN2. Compared with the levels in the control cells, 500 µM H_2_O_2_ upregulated optic atrophy protein 1 (OPA1) and dynamin-related protein 1 (DRP1). However, chrysoeriol pre-treatment increased OPA1 and decreased DRP1 levels compared with the levels in the H_2_O_2_-alone-treated group. Additionally, DRP1 Ser 616 phosphorylation was substantially reduced upon chrysoeriol pre-treatment compared with H_2_O_2_-alone treatment (Figure 5B). All expression levels were normalized to those of GAPDH. Together, our findings demonstrate that chrysoeriol may balance mitochondrial dynamics in ARPE-19 cells.

### 2.5. Chrysoeriol-Mediated Activation of p38 and Mitochondrial-Related Genes

The p38 mitogen-activated protein kinase (MAPK) activation is related to redox signaling transduction via stimulation of stress responses, including mitochondrial dysfunction [27,28,29,30,31]. Therefore, the protective antioxidant effect of chrysoeriol on the p38 MAPK pathway was determined in ARPE-19 cells. Western blot analysis results reveal that the total intracellular p38 level was largely unaffected; however, the activated form of p38 (pp38) was significantly upregulated in the H_2_O_2_-treated group. In contrast, when ARPE-19 cells were co-exposed to chrysoeriol and H_2_O_2_, the pp38 protein level was substantially decreased compared with that of the H_2_O_2_ control (Figure 6A,B). Next, we sought to determine whether MAPK inhibitors mediate the phosphorylation of p38 and the protective effects of chrysoeriol. Cell viability was increased in the cells pre-treated with chrysoeriol compared with H_2_O_2_*-*alone-treated cells. The antioxidant effects of chrysoeriol were not affected by the other inhibitors (PD 98059; MEK/ERK pathway inhibitor, SP600125; c-Jun N-terminal kinase (JNK) inhibitor, and LY 294002; phosphatidylinositol 3-kinase (PI3K) inhibitor). However, treatment with SB203580 (p38 inhibitor) significantly decreased cell viability in the chrysoeriol pre-treatment group (Figure 6C). To characterize the role of p38 activation in Nrf2 signaling regulation, ARPE-19 cells treated with a p38 inhibitor were also assessed by Western blot analysis. The chrysoeriol and H_2_O_2_ co-treated group exhibited an upregulation in Nrf2 and HO-1 expression, whereas the SB203580-treated group exhibited a decrease in Nrf2 and HO-1 expression (Figure 6D). The protein expression levels in Figure 6D were normalized to those of GAPDH (Figure 6E). SB203580 significantly blocked chrysoeriol-induced antioxidant molecules such as Nrf2 and HO-1 compared to the chrysoeriol and H_2_O_2_ co-treatment group. These findings indicate that the activation of the p38 pathway mediates the antioxidant effects of chrysoeriol and the activation of Nrf2 signaling in H_2_O_2_-induced damaged ARPE-19 cells.

### 2.6. Chrysoeriol Protected ARPE-19 Cells from Sodium Iodate-Induced Oxidative Damage

Sodium iodate (NaIO_3_) is an oxidizing agent that promotes ROS generation and is specifically toxic to RPE cells by inducing mitochondrial dysfunction [32,33,34,35]. To investigate this effect further, cells were exposed to NaIO_3_. A NaIO_3_ concentration of 20 mM significantly reduced cell viability by approximately 50%, whereas chrysoeriol pre-treatment significantly suppressed the NaIO_3_-induced reduction in ARPE-19 cell viability (Figure 7A). As shown in Figure 7B, compared with the control group, 20 mM NaIO_3_ caused a significant increase in the fluorescence intensities of H_2_DCFDA (B) and DHE (C) in the ARPE-19 cells. However, pre-treatment with chrysoeriol in the ARPE-19 cells substantially suppressed these increases compared with the levels in NaIO_3_ treatment alone (Figure 7B,C). When exposed to 20 mM NaIO_3_, the cells exhibited a reduction in MMP, whereas pre-treatment with chrysoeriol for 2 h suppressed this effect compared with the cells exposed to NaIO_3_ alone (Figure 7D). Together, our findings suggest that chrysoeriol could effectively protect RPE cells against pathological oxidative damage.

## 3. Discussion

Age is a key risk factor for the development of AMD, which is largely caused by oxidative stress resulting from elevated ROS levels. AMD is characterized by abnormal RPE cell layers, whereby superimposing foveal photoreceptor dysfunction ensues. Additionally, substantial oxidative stress build-up may fully disrupt the antioxidant systems and result in irreversible retinal damage [2,36]. Here, we investigated the protective effects of chrysoeriol (a flavonoid compound) against oxidative damage in human RPE cells. At the mitochondrial level, chrysoeriol exerts its protective effects via MMP mediation and its related effector genes, including OXPHOS genes, mitochondrial process genes, and mitochondrial DNA replication and transcription genes. Notably, chrysoeriol may prevent oxidative damage by increasing the expression of antioxidant enzymes, mainly HO-1 and SOD2, via the upregulation of Nrf2 and mitochondrial molecules in ARPE-19 cells. Furthermore, we confirmed that the p38 signaling pathway was involved in chrysoeriol-mediated RPE cell death. Therefore, our findings highlight the potential therapeutic applicability of chrysoeriol to prevent or treat AMD, a disease initiated by cell death caused by oxidative stress and RPE dysfunction.

Excessive intracellular ROS has been linked to RPE cell oxidative damage and dysfunction [37]. Therefore, decreasing intracellular ROS may guard the RPE against oxidative damage [5,38]. Our study reveals that chrysoeriol markedly diminished H_2_O_2_-induced intracellular ROS levels in RPE cells, as demonstrated by our DCFDA and DHE assays. Next, we characterized the expression of major antioxidant enzymes in ARPE-19 cells, including SOD1, SOD2, catalase, glutathione peroxidase (GPx), HO-1, and NQO-1. Pre-incubation with chrysoeriol increased overall antioxidant-related genes (SOD1, SOD2, catalase, GPx, HO-1, and NQO-1) compared to the H_2_O_2_ group. Additionally, chrysoeriol markedly increased SOD2, HO-1, and NQO-1 expression levels in H_2_O_2_-exposed ARPE-19 cells. These findings indicate that chrysoeriol may possess the ability to scavenge oxygen free radicals, thereby indirectly combating oxidative stress. Numerous studies have reported that Nrf2 activation is a major regulator of several antioxidant and detoxification genes in RPE cells, including downstream targets of Nrf2 [39,40,41,42,43,44]. Our study suggests that chrysoeriol activates Nrf2 and increases HO-1 and NQO-1 activity in H_2_O_2_-exposed cells, as demonstrated by RT-qPCR and Western blot analyses.

Mitochondrial dysfunction is a key factor that leads to AMD pathological changes, including MMP (∆Ψm) reduction and mitochondrial DNA damage [45,46]. Additionally, previous studies have reported an overall reduction in mitochondria numbers in the RPE of elderly individuals, which was even more severe in AMD patients [47,48,49]. In this study, we found that pre-treatment with chrysoeriol significantly increased the MMP compared to H_2_O_2_ treatment alone. Next, we analyzed mitochondrial-related gene expression, and our results demonstrate that chrysoeriol significantly upregulated TFAM, POLG, ATP5O, COX4I1, COX5B, NDUFB5 MFN1, and MFN2. Mitochondria continuously undergo fission and fusion processes to maintain mitochondrial function. Mitochondrial fission and fusion enable the recycling of damaged elements via the segregation of injured organelles and the exchange of materials with healthy mitochondria [50,51,52]. DRP1 is a key mediator of mitochondrial fission, whereas OPA1 is pivotal for mitochondrial membrane fusion and maintaining proper mitochondrial cristae architecture [50,53,54]. Therefore, we investigated the expression of crucial mitochondrial process genes including TOM20, MFN2, OPA1, and DRP1. Our results indicated that chrysoeriol significantly increased TOM20, MFN2, and OPA1 and decreased DRP1, whereas H_2_O_2_ induced DRP1 activation. Thus, chrysoeriol may improve mitochondrial function and biogenesis, thereby alleviating H_2_O_2_-induced oxidative stress.

MAPKs, ERK1/2, p38, and JNKs, all of which are well-characterized mitogen-activated protein kinases, are known to be activated by oxidative stress and are involved in cell growth and death [55,56]. Increasing evidence suggests that regulating the MAPK signaling pathway, particularly p38 activation, is critical to protect cells from ROS injury and cellular death [57]. Here, H_2_O_2_-induced oxidative stress significantly promoted p38 phosphorylation [58]. Compared with the H_2_O_2_ group, p38 phosphorylation was lower in the chrysoeriol group, suggesting that chrysoeriol exerts its antioxidant effect through the modulation of the ROS-mediated p38 MAPK signaling pathway. Next, we confirmed that chrysoeriol treatment decreased p38 phosphorylation, whereas pre-treatment with a p38 inhibitor and chrysoeriol attenuated the cytoprotective effect of chrysoeriol against H_2_O_2_ in ARPE-19 cells. The p38 pathway negatively regulated the chrysoeriol-induced Nrf2/HO-1 expression, and treatment with a p38 inhibitor reduced the chrysoeriol-induced antioxidant protein expression. Recent studies have reported that p38 induces mitochondrial fragmentation by mediating DRP1. It has also been reported that PKCδ-activated p38 MAPK directly phosphorylates DRP1 to induce its mitochondrial translocation and subsequent mitochondrial fission. Based on our results, inhibition of p38 MAPK prevents mitochondrial dysfunction via inhibition of DRP1 phosphorylation and activation of Nrf2/HO-1; thus, our study demonstrates that chrysoeriol could lead to improved mitochondrial function and biogenesis against H_2_O_2_-induced oxidative stress in ARPE-19 cells. Furthermore, we demonstrate that chrysoeriol not only exerts antioxidant effects in H_2_O_2_-exposed cells but also in cells exposed to NaIO_3_, an RPE-specific oxidizing agent that induces mitochondrial dysfunction.

In conclusion, our study demonstrates the protective effect of chrysoeriol against oxidative damage to ARPE-19 cells in experimental conditions that were meant to mimic AMD pathological development. Chrysoeriol exhibited potent antioxidant effects, as observed by increased cell survival, increased antioxidant enzyme expression, decreased ROS accumulation, increased OXPHOS, and increased mitochondrial-related gene expression, leading to increased MMP and mitochondrial function via the modulation of mitochondrial gene expression. The mechanisms by which chrysoeriol exerted these effects included the regulation of mitochondrial quality control molecules (e.g., DRP1 and OPA1) via the modulation of p38/ Nrf2/HO-1 signaling, as well as an increase in antioxidant molecules. Our study is the first to demonstrate the potential therapeutic applicability of chrysoeriol for the prevention of dry AMD. However, further studies are needed to determine the physiological function and biological efficacy of chrysoeriol in both primary human RPE cells (or at least fully differentiated ARPE-19 cell models) and in vivo models.

## 4. Materials and Methods

### 4.1. Materials

The ARPE-19 human retinal pigment epithelial cell line was purchased from ATCC (Manassas, VA, USA). Dulbecco’s modified Eagle’s medium, Nutrient Mixture F-12 media (DMEM/F12), fetal bovine serum (FBS), penicillin/streptomycin, and 2,7-dichlorofluorescein diacetate (H_2_DCF-DA) were purchased from Thermo Fisher Scientific (Wilmington, DE, USA). Hydrogen peroxide, dihydroethidium (DHE), JC-10 assay kits, *N*-acetyl-cysteine (NAC), and carbonyl cyanide m-chlorophenyl hydrazone (CCCP) were purchased from Sigma-Aldrich (St. Louis, MO, USA, owned by Merck KGA). The Cell Counting Kit-8 (CCK-8) was purchased from Dojindo Molecular Technologies (Kumamoto, Japan). PD98059, SP60025, SB202580, and LY294002 were purchased from Cayman Chemical (Ann Arbor, MI, USA). Antibodies against Nrf-2, NQO-1, and HO-1 were purchased from Abcam (Cambridge, UK). p38, pp38, TOM20, MFN2, DRP1, OPA1, and p-DRP1 antibodies were obtained from Cell Signaling Technology (Beverly, MA, USA).

### 4.2. Preparation of Chrisoeriol

#### 4.2.1. The Source of Chrysoeriol

We cultivated perilla (cultivar name Anyu, Perilla frutescens) seeds in an experimental field at the National Institute of Crop Science (NICS) in the Rural Development Administration (RDA), Jeonbuk, Korea, 2019. After harvesting, perilla seeds were immediately freeze-dried and stored at −40 °C.

#### 4.2.2. Purification Protocol and Purity Analysis of Chrysoeriol

The perilla seed contains approximately 35–40% oil (saturated and unsaturated). To remove the crude oil effectively, chrysoeriol was extracted with n-hexane twice. The resulting residue was extracted with 70% methanol at room temperature and then filtered through a filter paper. The filtrate residue was concentrated and evaporated to obtain the brown perilla seed extract. This extract was subjected to column chromatography on a silica gel (15 × 40 cm, 230–400 mesh, 370 g) by using acetone-di-chloromethane mixtures [1:30, 25:1, 20:1, 10:1, 5:1, 2:1, 1:1, and 1:2] to yield 6 fractions (A–F). Fraction B was re-chromatographed via silica gel column chromatography (5.5 × 40 cm, 230–400 mesh, 115 g) by using an acetone-ethylacetated gradient (15:1→1:4) to obtain chrysoeriol (158 mg). The isolated chrysoeriol was dissolved in methanol to 1.0 mg/mL and analyzed via ultra-high performance liquid chromatography (UHPLC)-photodiode array detection (PDA). The analysis was conducted using a reverse-phase UHPLC (Dionex Ultimate 3000, Thermo Scientific, Waltham, MA, USA) equipped with a Acclaim^TM^ RSLC Polar Advantage II (2.2 μm 120Å 2.1 × 150 mm) column. The mobile phase was 0.1% acetic acid in water (A) or 0.1% acetic acid in methanol (B). The solvent flow rate was 0.6 mL/min, and the column temperature was set at 40 ·C. The gradient was performed as follows: 0–1 min, 10% B; 2 min, 20% B; 2 min, 30% B; 3 min, 50% B; 3 min, 70% B; 5 min, 100% B; 5 min. Following injection of 1 μL of sample, chrysoeriol was eluted. The NMR (nuclear magnetic resonance) analysis was performed with ^1^H and ^13^C by using a Bruker AM 500 spectrometer [59]. The standard of chrysoeriol was purchased from ChemFaces (CFN 98785, Wuhan, China). The purity analysis data are provided in the Appendix A.

### 4.3. Cell Culture and Cell Viability Assay

ARPE-19 cells were routinely maintained in DMEM/F12 media supplemented with 10% FBS and 1% penicillin/streptomycin at 37 °C in a 5% CO_2_ atmosphere. Cell viability was assessed using the CCK-8 assay according to the manufacturer’s instructions. Each experiment was repeated three times with triplicate samples.

### 4.4. ROS Measurement

Intracellular ROS levels were examined using H_2_DCF-DA or DHE according to the manufacturer’s instructions. The fluorescence intensity was measured using a fluorescence plate reader (Bio-Tek) at Ex/Em = 495/527 nm for H_2_DCF-DA and Ex/Em = 535/610 nm for DHE. Afterward, H_2_DCFDA-stained cell images were obtained using an IX51 fluorescent microscope coupled with a DP microscope camera controller (Olympus Optical, Japan).

### 4.5. Mitochondrial Membrane Potential (MMP) Assay

The MMP assay was conducted using the JC-10 MMP assay kit (Merck, St. Louis, MO, USA) according to the manufacturer’s instructions. In brief, ARPE-19 cells (5 × 10^3^ cells/well) seeded in a 96-well transparent-bottom black plate (Eppendorf Ltd., Germany) were treated as described above. After 24 h, the JC-10 dye solution (JC-10 and assay buffer A 1:100 v/v) was added (50 µL/well) to the control and treated cells. Following the treatment, the plate was incubated in dark conditions for 30 min. Afterward, assay buffer B (50 µL/well) was added, and the fluorescence intensity was measured at 490/525 nm (red) and 540/590 nm (green) using a multimode plate reader (Bio-Tek). The red/green fluorescence intensity ratio was used to determine the MMP. Carbonyl cyanide m-chlorophenyl hydrazone (CCCP) and *N*-acetyl-cysteine (NAC) were used as negative and antioxidant controls, respectively.

### 4.6. RNA Collection and Quantitative PCR

Total RNA was collected using the TRIzol reagent (Thermo Fisher Scientific) and resuspended in RNAse-free water. Reverse transcription was performed with 1 μg of RNA to produce complementary DNA (cDNA) using the SensiFast cDNA synthesis kit (Bioline, London, UK). RT-qPCR was performed using 3 μL of cDNA template and the Power SYBR Green Master Mix (Thermo Fisher Scientific) in a StepOnePlus thermocycler (Applied system, USA). The primer sequences for GAPDH, SOD1, SOD2, CAT, GPx, HO-1, NQO-1, Nrf-2, TFBM, POLG, ATP5O, Cox4I1, COX5B, NDUFB, FIS1, MFN1, and MFN2 are listed in Table 1.

### 4.7. Western Blotting

Cells were lysed in RIPA buffer with protease inhibitors. An equal amount of protein was separated by 10% SDS-PAGE gel and transferred to PVDF membranes (Millipore, Bedford, MA, USA). The membrane was blocked with 5% skim milk in TBST, incubated with a primary antibody overnight at 4 °C, and then allowed to react with the secondary antibodies for 2 h at room temperature. Protein expression levels were evaluated with an enhanced chemiluminescence kit (Bio-Rad Laboratories, Inc., Hercules, CA, USA).

### 4.8. Statistics

Statistical analyses were performed using the GraphPad Prism 5 software (GraphPad Software Inc., La Jolla, CA, USA). All data were reported as mean ± SD. Student’s t-test was used to calculate pairwise statistical significance between two groups. Statistical significance was denoted as follows: ns *p* >0 05; *, # *p* < 0 05; **, ## *p* <0 01; ***, ### *p* < 0 001.

## Figures and Tables

**Figure 1 molecules-26-00313-f001:**
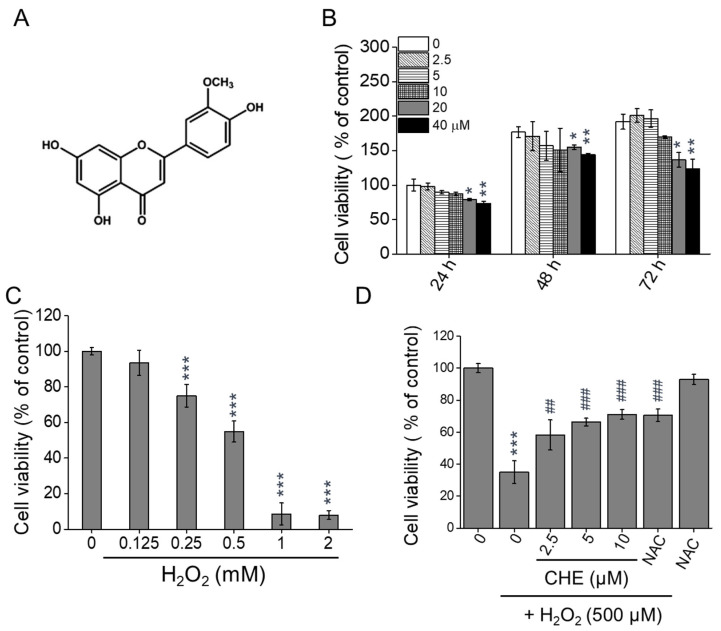
Protective effects of chrysoeriol (CHE) against H_2_O_2_-induced cytotoxicity in ARPE-19 cells. (**A**) Schematic drawing of the structures of the CHE complexes. (**B**) Cells were treated with CHE (2.5–40 μM) or 0.1% DMSO (vehicle control) for 24–72 h, and the cell viability was measured. (**C**) Cells were treated with H_2_O_2_ (0.125–2 mM) for 24 h, and the cell viability was measured. (**D**) Cells were pre-treated with CHE at the indicated concentrations or 0.1% DMSO (vehicle control) for 2 h and then incubated with or without 500 μM H_2_O_2_ for 24 h. *N*-acetyl-cysteine (NAC, 4 mM) was used as an antioxidant control. Cell viability was measured via the CCK-8 assay (B, C, and D). * *p* < 0.05, ** *p* < 0.01, *** *p* < 0.001, versus the control group; ## *p* < 0.01, ### *p* < 0.001, versus the H_2_O_2_-treated group.

**Figure 2 molecules-26-00313-f002:**
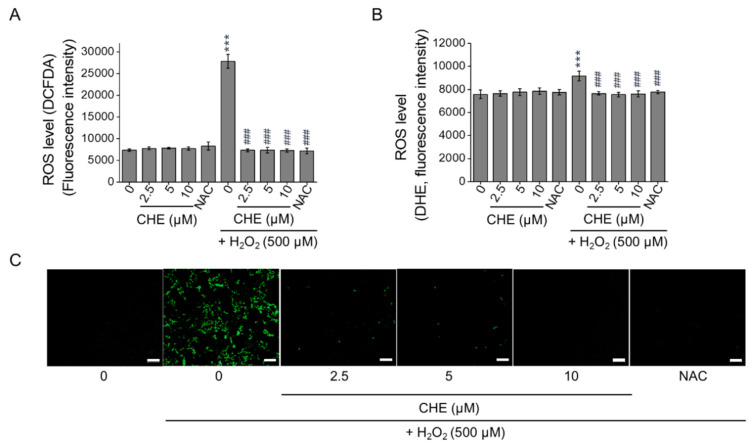
Protective effects of CHE against H_2_O_2_-induced ROS generation in ARPE-19 cells. Cells were pre-treated with 2.5, 5, or 10 μM CHE or 0.1% DMSO (vehicle control) for 2 h and then incubated with or without 500 μM H_2_O_2_ for an additional 24 h. Afterward, ROS levels were measured by using DCFDA (**A**) or DHE (**B**). Representative images of the cells evaluated by DCFDA staining. (**C**) *N*-acetyl-cysteine (NAC, 4 mM) was used as an antioxidant control. *** *p* < 0.001, versus the control group; ### *p* < 0.001, versus the H_2_O_2_-treated group. Scale bar: 100 μm.

**Figure 3 molecules-26-00313-f003:**
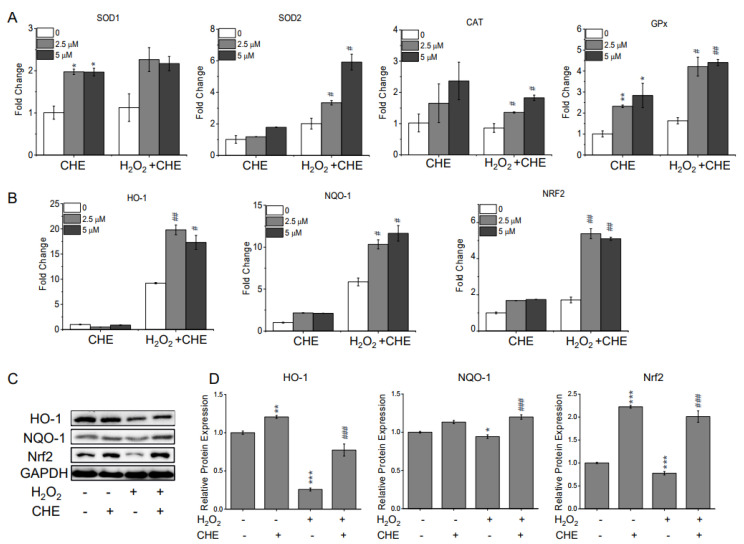
Involvement of the HO-1/ Nrf2 axis in the antioxidant effects of CHE. Cells were pre-treated with 2.5 or 5 μM CHE or 0.1% DMSO (vehicle control) for 2 h and then incubated with or without 500 μM H_2_O_2_ for an additional 24 h. Afterwards, the cells were harvested to extract RNA and protein. Expression analysis of antioxidant marker genes via RT-qPCR. All the gene expression data were analyzed using Student’s *t*-test (**A**,**B**). (**C**) Cells were pre-treated with 5 μM CHE or 0.1% DMSO for 2 h and then incubated with or without 500 μM H_2_O_2_ for an additional 24 h. HO-1, NQO-1, and Nrf2 protein levels were quantified via Western blot analysis. (D) HO-1, NQO-1, or Nrf2 expression level was plotted relative to the control level. * *p* < 0.05, ** *p* < 0.01, *** *p* < 0.001, versus the control group; # *p* < 0.05, ## *p* < 0.01, ### *p* < 0.01, versus the H_2_O_2_-treated group.

**Figure 4 molecules-26-00313-f004:**
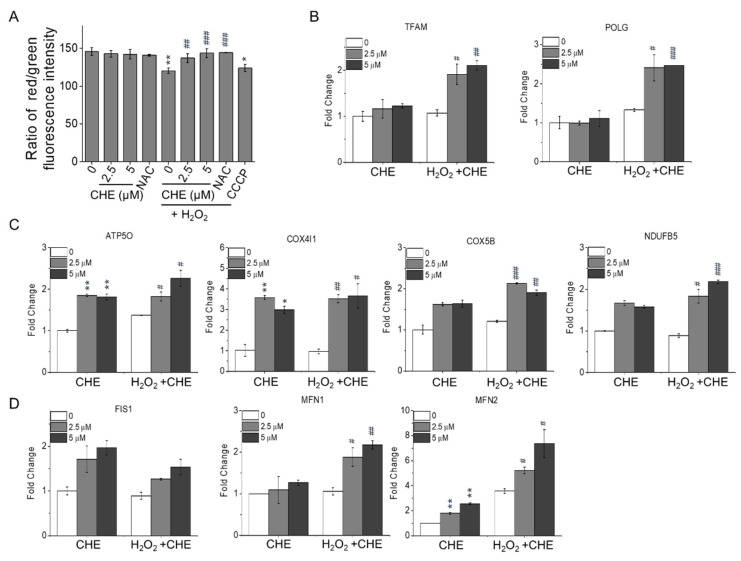
(**A**) CHE attenuated H_2_O_2_-induced mitochondrial membrane potential loss. After pre-treatment with 2.5 or 5 μM CHE for 2 h, ARPE-19 cells were incubated with or without 500 μM H_2_O_2_ for an additional 24 h. Red/green fluorescence intensity was quantified via the JC-10 assay (**A**). Carbonyl cyanide m-chlorophenyl hydrazone (CCCP, 40 μM) and *N*-acetyl-cysteine (NAC, 4 mM) were used as positive and antioxidant controls, respectively. CHE upregulated the expression of replication genes (**B**), oxidative phosphorylation (OXPHOS) target genes (**C**), and mitochondrial dynamics genes (**D**) in ARPE-19 cells. Relative gene expression of the replication genes (**B**), OXPHOS genes (**C**), and mitochondrial dynamic genes (**D**) in ARPE-19 cells treated for 2 h with 2.5 or 5 μM CHE and then incubated with or without 500 μM H_2_O_2_ for an additional 24 h. * *p* < 0.05, ** *p* < 0.01, versus control group; # *p* < 0.05, ## *p* < 0.01, ### *p* < 0.001, versus the H_2_O_2_-treated group.

**Figure 5 molecules-26-00313-f005:**
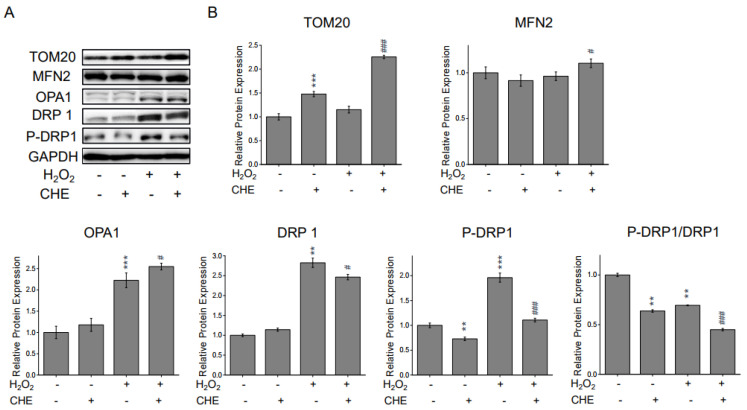
CHE increased TOM20, OPA1, and MFN2 protein expression and decreased DRP1 and P-DRP1 protein expression. The cells were pre-treated with 5 μM CHE or 0.1% DMSO (vehicle control) for 2 h and then incubated with or without 500 μM H_2_O_2_ for an additional 24 h before being subjected to Western blot analysis. (**A**) Compared with ARPE-19 cells treated with H_2_O_2_-only and pre-treated with CHE and H_2_O_2_, CHE increased TOM20, MFN2, and OPA1 protein expression and decreased DRP1 and P-DRP1 protein expression. Protein levels of TOM20, MFN2, OPA1, DRP1, and P-DRP1 were quantified via Western blotting. (**B**) Representative Western blot results. ** *p* < 0.01, *** *p* < 0.001, versus control group; # *p* < 0.05, ### *p* < 0.001, versus the H_2_O_2_-treated group.

**Figure 6 molecules-26-00313-f006:**
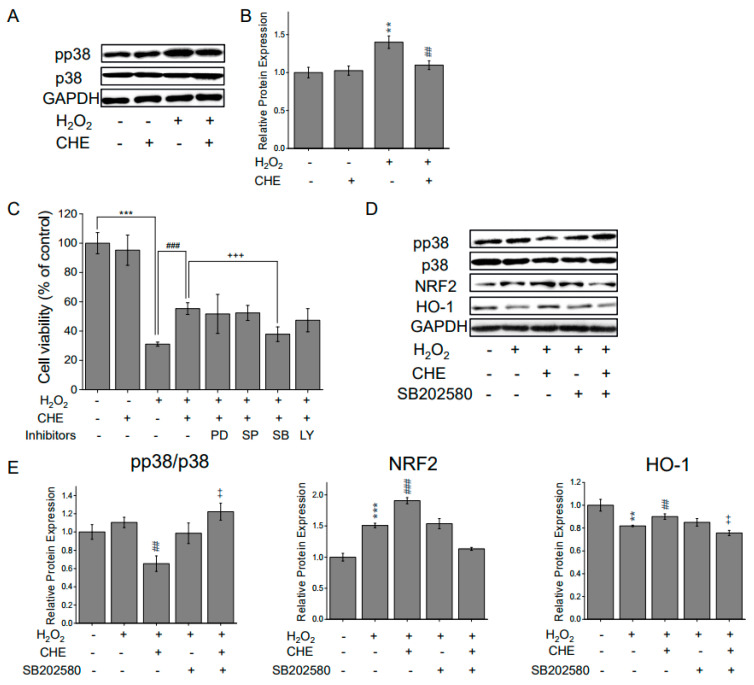
CHE inhibits the activation of p38 MAPK during H_2_O_2_-induced oxidative stress, and p38 MAPK inhibition mediates CHE-induced Nrf2 activation. (**A**) ARPE-19 cells were treated with or without 5 μM CHE for 2 h and then treated with H_2_O_2_ for 30 min. (**C**) Cells were pre-treated with a specific kinase inhibitor (PD98059 (ERK), SP600125 (JNK), SB203580 (p38), or LY 294002 (PI3K)) at 10 μM for 2 h, 5 μM CHE for another 2 h, and then followed by treatment with 500 μM H_2_O_2_ for 24 h. Cell viability was quantified via the CCK-8 assay. (**D**) ARPE-19 cells were pre-treated with a p38 inhibitor at 10 μM for 2 h, 5 μM CHE for another 2 h, and then treated with 500 μM H_2_O_2_ for 24 h. The protein levels of p38, pp38, HO-1, NQO-1, and Nrf2 were assessed via Western blotting. (**B**,**E**) The quantitative analysis of the p38, pp38, HO-1, NQO-1, and Nrf2 levels was performed via densitometric measurements relative to the control. ** *p* < 0.01, *** *p* < 0.001, versus control group; ## *p* < 0.01, ### *p* < 0.001, versus the H_2_O_2_-treated group; ++ *p* < 0.01, +++ *p* < 0.001, versus the H_2_O_2_ and chrysoeriol-treated group.

**Figure 7 molecules-26-00313-f007:**
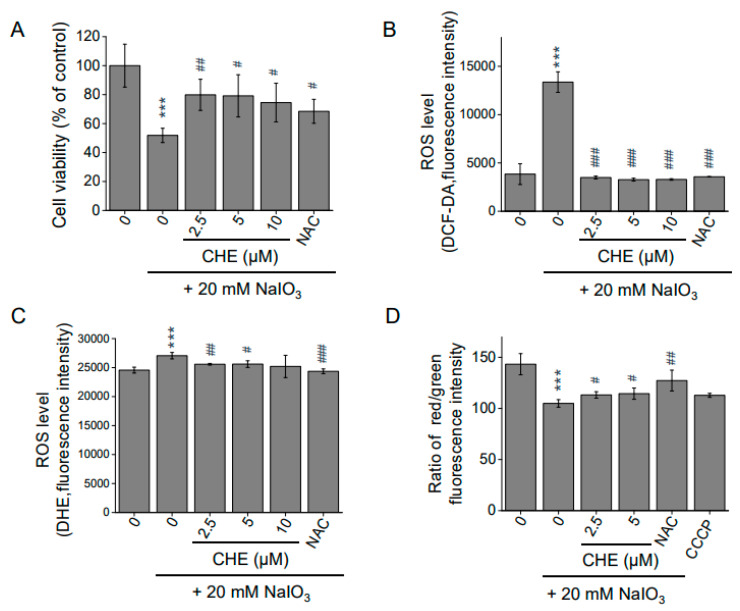
Protective effects of CHE against NaIO_3_-induced cell damage in ARPE-19 cells. (**A**) Cells were pre-treated with CHE at the indicated concentrations or 0.1% DMSO (vehicle control) for 2 h and then incubated with or without 20 mM of NaIO_3_ for an additional 24 h. Cell viability was measured via the CCK-8 assay. ROS levels were measured by the DCFDA (**B**) and DHE (**C**) assays. *N*-acetyl-cysteine (NAC, 4 mM) was used as an antioxidant control. (**D**) CHE attenuated NaIO_3_-induced loss of mitochondrial membrane potential. After pre-treatment with 2.5 or 5 μM CHE for 2 h, ARPE-19 cells were incubated with or without 20 mM NaIO_3_ for an additional 24 h. Red/green fluorescence intensity was quantified via the JC-10 assay (A). Carbonyl cyanide m-chlorophenyl hydrazone (CCCP, 40 μM) and *N*-acetyl-cysteine (NAC, 4 mM) were used as positive and antioxidant controls, respectively. *** *p* < 0.001, versus control group; # *p* < 0.05, ## *p* < 0.01, ### *p* < 0.001 versus the NaIO_3_-treated group.

**Table 1 molecules-26-00313-t001:** Primer sequences.

Target Gene	Forward Sequence (5′-3′)	Reverse Sequence (5′-3′)
ATP5O	CGCTATGCCACAGCTCTTTA	AAGGCAGAAACGACTCCTTG
COX4I1	GGCATTGAAGGAGAAGGAGA	TCATGTCCAGCATCCTCTTG
COX5B	GAGGTGGTGTTCCCACTGAT	CAGACGACGCTGGTATTGTC
NDUFB5	CTTCCTCACTCGTGGCTTTC	TCTGGGACATAGCCTTCTGG
FIS1	GACATCCGTAAAGGCATCGT	ACAGCAAGTCCGATGAGTCC
MFN1	TGCCCTCTTGAGAGATGACC	TCTTTCCATGTGCTGTCTGC
MFN2	ATGCATCCCCACTTAAGCAC	GCAGAACTTTGTCCCAGAGC
TFAM	TAAGACTGCAAGCAGCGAAG	TTCTCAGTTTCCCAGGTGCT
POLG	TGCAGTGAGGAGGAGGAGTT	CCCAGGTAAGTGCCATGAGT
SOD1	GAAGGTGTGGGGAAGCATTA	CTTTGCCCAAGTCATCTGCT
SOD2	AAACCTCAGCCCTAACGGTG	GCCTGTTGTTCCTTGCAGTG
CAT	GATAGCCTTCGACCCAAGCA	AGAAGGCTGTTGTTCCGGAG
GPX1	AGTCGGTGTATGCCTTCTCG	CAAACTGGTTGCACGGGAAG
HO-1	AGTCTTCGCCCCTGTCTACT	GCTTGAACTTGGTGGCACTG
NQO-1	AAAGGACCCTTCCGGAGTAA	CGTTTCTTCCATCCTTCCAG
NRF2	GCGACGGAAAGAGTATGAGC	ACGTAGCCGAAGAAACCTCA
GAPDH	ACCCAGAAGACTGTGGATGG	TTCTAGACGGCAGGTCAGGT

## Data Availability

Data available in a publicly accessible repository.

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
