# Peer review of "Antioxidative Effects of Chrysoeriol via Activation of the Nrf2 Signaling Pathway and Modulation of Mitochondrial Function"

_molecules, 2021, doi:10.3390/molecules26020313_

Round 1

Reviewer 1 Report

In the paper “Anti-Oxidative Effects of Chrysoeriol via Activation of the Nrf2 signaling Pathway and Modulation of Mitochondrial Function” the authors confirmed the protective role of Chrysoeriol against oxidative stress in ARPE19 cells and analysed the signal transduction pathway involved in the protection.

Although, of potential interest, the paper, in my opinion, is not suitable for publication in Molecules in this version.

Major concerns :

The description of the experiments is not always clear in particular.

  • Fig 1B Statistical analysis is needed.
  • How long after H2O2 treatment cells were harvested to extract RNA to analyse the expression of antioxidant enzymes and mithocondrial related genes?
  • Fig 3C: which concentration of CHE has been used?
  • How long after H2O2 treatment cells were harvested to extract proteins for experiments in Fig 3C and Fig 5?
  • Rows 198-199 should be rewritten clarifying the signal transduction pathway blocked by the different inhibitors.
  • There is an incongruence between Figure 6C and figure legend: in the figure is reported that cells were pre-treated with CHE and different inhibitors the legend describe that “Cells were pre-treated with 5 μM CHE or 10 μM of inhibitors”

Furthermore, the authors should investigate whether chrysoeriol improves mitochondrial function and biogenesis blocking the activation of p38 pathway or through an independent pathway

Author Response

Response letter

Dear Ema Lupsic:

On behalf of all the authors, we would like to thank you for allowing us to resubmit our manuscript. We appreciate the valuable comments and suggestions of the reviewers. Our point-by-point responses (in blue) to these comments are below. We have modified the manuscript according to the suggestions of the reviewers, and these inputs have significantly improved the manuscript. We hope that the revised manuscript is suitable for publication in Molecules.

Again, we would like to emphasize that we are very grateful for all the comments of the reviewers, which tremendously improved the clarity of our results.

Sincerely,

Dae Yu Kim, PhD

Department of Electrical Engineering, College of Engineering,

Inha Research Institute for Aerospace Medicine,

Inha University, Incheon 22212, South Korea

Reviewer 2 Report

RPE cell dysfunction caused by excessive oxidative damage is involved in AMD development and progression leading to visual impairment in elderly people. Currently, effective treatment for dry AMD is not yet available and the submitted manuscript highlights the potential therapeutic applicability of chrysoeriol to treat and prevent AMD.

This study demonstrates that chrysoeriol exerts a protective role against H2O2-induced oxidative stress and mitochondrial dysfunction via the upregulation of the antioxidant-associated genes and the regulation of mitochondrial dynamics molecules in ARPE-19 cells.

This is a detailed, well-designed, performed and presented study. However, before accepting this manuscript for publication the authors are requested to address the following comments.

  1. Indicate the source of Chrysoeriol, provide brief purification protocol, analysis methods and degree of purity.
  2. Please add the following controls – effect of NAC alone (without H2O2) on:
    1. Cell viability in Fig. 1D
    2. ROS in Fig. 2(A-C)
    3. MMP in Fig. 4A
    4. the tested parameters in Fig. 7(A-D).
    5.  
  3. Please add the following controls – effect of CHE alone (without H2O2) on:
    1. ROS in Fig. 2(A-C)
    2. the tested parameters in Fig. 6(D-E)
    3. the tested parameters in Fig. 7(A-D)
    4. CCCP treated cells in Fig. 4A

Fig. 4 shows that CHO treatment alone significantly increased ATP5O, COX4I1, MNF2 but not TFAM, POLG, COX5B, NDUFB5 and MFN1 while combined treatment of CHO plus H2O2 increased all indicated parameters – please discuss these differences.

Author Response

(The authors gave the same response as above.)

Round 2

Reviewer 1 Report

The manuscript significantly improved in this new version.

The paper is now suitable for publication in Molecules.

Reviewer 2 Report

The authors properly replayed to my comments and the manuscript is acceptable for publication in its present form.